# Gluten-Free Diet: Nutritional Strategies to Improve Eating Habits in Children with Celiac Disease: A Prospective, Single-arm Intervention Study

**DOI:** 10.3390/nu13041108

**Published:** 2021-03-28

**Authors:** Marta Suárez-González, Carlos Bousoño-García, Santiago Jiménez-Treviño, Juan José Díaz-Martín

**Affiliations:** Pediatric Gastroenterology and Nutrition Unit, Central University Hospital of Asturias, Oviedo, Asturias. Av. Roma, s/n, 33011 Oviedo, Asturias, Spain; ringerbou@yahoo.es (C.B.-G.); principevegeta@hotmail.com (S.J.-T.); juanjo.diazmartin@gmail.com (J.J.D.-M.)

**Keywords:** dietitian, nutritional management, healthy eating, gluten-free diet, dietary therapy

## Abstract

Background: Elimination of gluten-containing cereals and consumption of ultra-processed gluten-free foods might cause an unbalanced diet, deficient in fiber and rich in sugar and fat, circumstances that may predispose celiac children to chronic constipation. Aim: to evaluate if counseling with a registered dietitian (RD) was capable of improving eating and bowel habits in a celiac pediatric population. Methods: Dietetic, lipid profile and stool modifications were analyzed, comparing baseline assessments with those twelve months after receiving heathy eating and nutrition education sessions. At both time points, 3-day food records, a bowel habit record and a lipid panel were conducted. Calculated relative intake of macro- and micro-nutrients were compared with current recommendations by the European Food Safety Authority (EFSA). Student’s paired *t*-test, McNemar test, Mandasky test and Pearson correlation tests were used. Results: Seventy-two subjects (58.3% girls) with a mean (standard deviation (SD)) age of 10.2 (3.4) years were included. Baseline diets were imbalanced in macronutrient composition. Significant improvements were observed in their compliance with dietary reference values (DRVs), where 50% of the subjects met fat requirements after the education and 67% and 49% with those of carbohydrates and fiber, respectively (*p* < 0.001). Celiac children decreased red meat and ultra-processed foods consumption (*p* < 0.001) and increased fruits and vegetables intake (*p* < 0.001), leading to a reduction in saturated fat (*p* < 0.001) and sugar intake (*p* < 0.001). Furthermore, 92% of the patients achieved a normal bowel habit, including absence of hard stools in 80% of children constipated at baseline (*p* < 0.001). Conclusions: RD-led nutrition education is able to improve eating patterns in children with celiac disease (CD).

## 1. Introduction

Celiac disease (CD) is an autoimmune enteropathy based on genetic susceptibility and triggered by the exposure of gluten [1]. Treatment consists of a lifelong gluten-free diet (GFD), including any meal made with this protein complex prolamins, present in the seeds of wheat, rye, barley and other closely similarly related grains [2] or in which gluten is present as a concealed component [3].

The exclusion of gluten-containing cereals, together with the consumption of poor nutritional quality, ultra-processed gluten-free foods, might contribute the GFD to induce a negative impact on health [4,5,6,7,8,9,10]. Several studies have observed that many celiac patients on GFD show diets deficient in fiber [4,7,8,9,10,11,12] and rich in sugar [6,13] and fat [4,5,6,7,8,9,14,15]. This is currently relevant considering that chronic constipation is a common and significant symptom in celiac patients [16] and that obesity is increasing among this population [17]. Furthermore, GFD is related with a higher risk of mineral and vitamin deficiencies [18], especially for iron [4,7,15,19,20], calcium [4,19,21,22,23], selenium, zinc, magnesium, vitamin B12, folate [7], vitamin D [4,18,23,24,25,26] and vitamin C [7,27].

Besides, low fiber is related to lower quality of health in CD patients [28], the consequences of chronic exposure to poor diet quality and sedentarism associated to atherosclerosis cardiovascular disease [29]. Therefore, for maintaining cardiovascular and overall health, strong evidence and broad consensus speak to minimizing consumption of added sugars, refined grains and avoiding industrial trans-fatty acids, while replacing them with plant- and animal-based whole food [30].

The objective of our study was to analyze the modifications in the diet and in the bowel habits of pediatric patients with CD after the implementation of a dietetic education program managed by an experienced registered dietitian (RD), focusing on achieving a healthy GFD that provides adequate amounts of nutrients. It should be rich in fruits and vegetables, whole grains, fish, legumes and nuts, but low in refined grains and simple sugars. We hypothesized that this intervention would help celiac pediatric patients to improve their dietary patterns to meet the requirement for protein, carbohydrate, fat, vitamins and minerals for healthy children and to prevent disease further in life as through diet. 

## 2. Materials and Methods

### 2.1. Participants

The study was conducted in the Pediatric Gastroenterology and Nutrition Unit of a tertiary hospital, from February 2018 to November 2019.

The inclusion criteria were as follows: children aged from 1 to 18 years with CD diagnosed according to the ESPGHAN 2012 criteria [31], followed in a pediatric gastroenterology unit and who had been following a strict GFD for at least 6 months prior to the inclusion in the study.

The exclusion criteria were: subjects diagnosed with gluten allergy, individuals who refused to sign the informed consent form or who abandoned the clinical follow-up and any who had received previous dietetic education from the RD at the clinic.

### 2.2. Nutritional Assessment

Blood lipid profile and stool information were collected together with a dietetic evaluation. The measurements of total cholesterol, HDL cholesterol, LDL cholesterol and triglycerides were obtained from medical records.

The bowel habit was obtained by asking the subjects their stool consistency according to the Bristol stool scale [32], which was shown graphically in the office. Based on the interpretation of the Bristol scale, type 1 and 2 stools were classified as hard stools, types 3 and 4 as normal stools and types 5, 6 and 7 as diarrheal stools.

Dietary evaluation was undertaken using a 3-day food record, which included all the meals consumed by the individual over a 3-day period, including one weekend day. Their caregivers weighed foods using a scale with a precision of 1 g and also used estimated household supported with a list of typical portion sizes of common food items. The survey was analyzed using a Spanish dietary analysis website (ODIMET) [33] and the related contributions of macro- and micro-nutrients were estimated. The findings were compared with dietary reference values (DRVs) for nutrient intake established by the European Food Safety Authority (EFSA) [34].

### 2.3. Nutrition Education

Children with CD and their caregivers received oral and written food and nutrition education. They were taught about healthy gluten-free eating and a healthy lifestyle through individualized nutrition advised based on the Healthy Eating Plate recommendations, adapted according to celiac needs, by the Harvard School of Public Health [35]. Written examples of age-specific sample diets were also provided.

Six months after the intervention, the subjects returned to the pediatric nutrition unit where the RD again collected the stool data. Further education was given to motivate and reinforce behaviors and to support previously established dietary modifications. They were also provided with a further 3-day food record for complementation by the next visit.

Twelve months after receiving nutrition counseling, the third and last visit with the RD took place. The participants submitted the second completed 3-day food record and underwent the same complete nutritional assessment performed at baseline.

### 2.4. Statistical Analysis

Statistical analysis was performed using R program, version 3.6.0 (R Foundation for Statistical Computing, Vienna, Austria) [36]. A descriptive analysis was performed, providing relative and absolute frequency distributions for qualitive variables and position and dispersion measures for quantitative variables. To evaluate changes after the nutrition intervention, a paired Student’s t-test was applied for quantitative variables and McNemar’s test for qualitative variables. To evaluate the homogeneity of distribution between paired samples and for qualitative variables with more than two levels, the Madansky test was used. A *p*-value < 0.05 was considered statistically significant. The linear relationship between quantitative variables was evaluated with Pearson or Spearman correlation tests, depending on whether the normality hypothesis was verified or not. Correlation coefficients were considered strong (above 0.7) moderate (0.4–0.6) or weak (0.2–0.4) [37].

## 3. Results

### 3.1. Subjects’ Characteristics

Seventy-two individuals (58.3% female) met the inclusion criteria. The mean (standard deviation (SD)) age of the participants was 10.2 (3.4) years and they were on a GFD for 2.5 (1.8) years. The nutritional state of the participants and changes in body composition from baseline to the end of the study has already been published elsewhere [38].

### 3.2. Dietary Intake

#### 3.2.1. Macronutrients

According to European DRVs, baseline diets were imbalanced in macronutrient composition for fat, carbohydrates, proteins and fiber. All participants followed a high-protein diet both before and after the nutrition education, but a significant decrease was observed [38]. After the dietitian intervention, the compliance with the recommendations showed a 117.4% improvement for total fat intake (*p* < 0.001), 45.7% for carbohydrates (*p* < 0.001) and in 58.1% (*p* < 0.001) for fiber (Figure 1). Significant improvements were also observed in saturated fat *mean (SD) –10.9 (10.14) g/day, *p* < 0.001) and sugar intake (mean (SD) –10.17 (26.74) g/day, *p* < 0.001).

#### 3.2.2. Micronutrients

According to the DRVs for minerals recommended by EFSA, we observed that all children on GFD met the requirements for phosphorus and almost all for potassium and selenium, both at the beginning and at the end of the study. Regarding calcium and iron intake, 40.3% and 52.8% respectively, met the recommendations in their initial diet, decreasing to 21% for calcium (*p* = 0.014) and 43.1% for iron, at the end of the study. Compliance with zinc and iodine recommendations by 54.2% and 1.4% respectively, remained constant. In contrast, the percentage of compliance with magnesium and copper recommendations increased by 66.7% and 63.6%, respectively. Compliance with sodium recommendations was decreased (Figure 2).

Regarding vitamin intake, a majority of patients met EFSA recommendations for vitamins B1, B3 and B6 at inclusion and at the follow-up of the study. A slight decrease in compliance with current recommendations was observed for vitamin B12, being statistically significant for vitamin E (*p* = 0.043). An increase was observed for the rest of the vitamins, including vitamin A, B2, folates, vitamin C and D (Figure 3).

#### 3.2.3. Foods

Taking into account animal-based food, a decrease in meat intake, especially red meat, and an increase in oily fish was observed. The consumption of milk decreased, while an increase was observed for natural unsweetened yogurt and soft cheese (Figure 4).

A clear decrease was observed in the intake of all ultra-processed products, highlighting processed meats such as cold cuts, sweetened yogurts and other sugary dairy desserts, cookies and other types of pastries (Figure 5).

An increase was observed in the amounts of healthy plant-based foods (Figure 6). The intake of other plant-based foods, classified in the group of gluten-free cereals and tubers, was also increased (Figure 7).

### 3.3. Stools’ Characteristics

According to the Bristol scale, percentage of participants with normal bowel showed a 43.3% increase (*p* < 0.001) at six months and a 53.3% increase (*p* < 0.001; Figure 8) one year after nutrition intervention.

### 3.4. Relationship of the Lipid Profile With Dietary Variables

Consumption of biscuits, saturated fat and total sugars was weakly positively associated with total blood cholesterol levels. Intake of processed meats, sugary soluble cocoa and saturated fat were weakly related to an increase in LDL cholesterol. A weakly negative association was observed between consumption of sugary breakfast cereals and skimmed yogurt and HDL cholesterol levels. Consumption of sugary breakfast cereals was weakly positively associated to triglyceride levels (Table 1). No correlation was observed between dietary cholesterol and blood cholesterol profile, for total (*p* = 0.928), LDL (*p* = 0.990) or HDL (*p* = 0.810).

## 4. Discussion

A GFD is currently the only available therapy for CD. This restriction implies the substitution of cereals with gluten and its derivatives by other foods with similar nutritional characteristics. If celiac children choose ultra-processed substitutes instead of traditional homemade meals, it could induce a negative impact on health [10]. Ultra-processed food consumption drives non-communicable diseases (NCDs) [39], such as obesity [40,41], diabetes [42], heart disease [43] and cancer [44]. In particular, added sugar is the prevalent, insidious and egregious component of ultra-processed food that drives that risk [45]. Nowadays, due to the caregiver’s lack of time, children end up eating ultra-processed and fast foods on a regular basis [46].

Furthermore, prior literature has shown that the diet of CD children is nutritionally less balanced than the diet of healthy children, richer in fat and poorer in fiber intake [47,48,49,50,51], highlighting the need for proper dietary counseling [52]. Forchielli et al. [53] underlined that protein intake in children’s diets, both before and after starting a GFD, more than doubled the recommended intakes. Gluten-free ultra-processed foods commonly have less protein content than their gluten-containing counterparts [49]; hence, the excessive protein intake observed in this population, as in our patients, is mainly a consequence of the consumption of animal protein [53]. These observations are worrisome since the consumption of red and processed meats has significant negative effects on health, and is related with higher risk of mortality from all causes [54].

These unbalanced eating patterns were also observed in our study. Baseline diets were high in fat and protein and low in starches and fiber. The present study showed that dietetic intervention based on the Healthy Eating Plate [55] resulted in improvements in these children’s dietary patters. Positive modifications included a significant decrease in the intake of processed foods, especially processed meat, sweetened yogurt, sweetened dairy desserts, processed cheese, pastries, biscuits, sweets, savory snacks, cocoa powder, bottled juices and soft drinks. Following Harvard Medical School recommendations to limit milk/dairy and red meat intake, the celiac patients also decreased the global consumption of dairy (at the expense of milk and sugary dairy products) and meats, as well as increases in the consumption of unsweetened plain yogurt, white fish, oily fish and eggs, were also demonstrated. Moreover, an increase of plant-based foods’ intake including fruits, vegetables, nuts, potatoes and rice was also achieved. In turn, after said established food education, they were able to make more whole wheat bread, dough and healthy pastries at home.

Knowing that constipation is associated with a low fiber intake [56], it was considered that the improvement achieved in the stool type of the children in the study was due to their increased fiber consumption. Specifically, it was achieved that practically all the patients had smooth and soft stools at the end of the study, with hard lumps disappearing in 80% of those who had them. Coinciding with published meta-analyses [57], we believe that adequate fiber intake should be part of a normal diet to prevent childhood constipation, so the celiac pediatric population should be encouraged to increase the plant-based foods intake (fruit, vegetables, nuts, legumes and whole grains).

GFD can also be deficient in micronutrients. In particular, it is characterized by low levels of vitamin B, folic acid, vitamin D, calcium, iron, zinc and magnesium [12,19,58]. Zuccotti et al. [59] observed that both celiac children and control subjects had insufficient intake of calcium, iron and magnesium according to the recommendations of the European Union Commission, as also confirmed by Turnbull et al. [60] in other studies in children.

Coinciding with these findings, when comparing the European recommendations with the diets of our patients, we also observed that it was deficient in magnesium in most cases, in iron in 47% and in calcium in 60%. This last observation is striking, although it coincides with the widely reported calcium deficiencies [4,19,21,22,23] in patients with CD who adhered to a GFD. Iron deficiencies in celiac patients with GFD were also observed by Mariani et al., [15] Thompson et al., [19] Martin et al., [7] Sue et al. [4] and Shepherd and Gibson [20].

Öhlund et al. [8] found that the intake of iron and calcium was higher in children with CD than in controls, but lower with respect to selenium, zinc and magnesium. In the specific case of selenium, the diet of our patients covered the requirements for this mineral, but the copper requirements were not met by most of the celiac children in the study.

Finally, we observed that the results obtained regarding sodium in the diet, which was not sufficient, was due to the fact that the salt added to the dishes was not included in the dietary analysis. This coincides with the low levels of iodine found, since in the western population, the main source of iodine is iodized salt [61].

Regarding vitamins, deficiencies in GFD were also observed, with vitamin B12, folate and vitamin D being the most affected. Hallert et al. [27] observed a lower vitamin B12 and folic acid intake in half of the celiac subjects studied with a GFD for 10 years compared to controls. Martin et al. [7] also reported a lower intake of vitamin B12, along with folic acid and vitamin C. Vitamin D deficiency has also been observed in subjects with CD and GFD [4,18,23,24,25,26]. However, Marger et al. [25] reported that the suboptimal level of vitamin D found at diagnosis resolved in half of the celiac population after one year with GFD. Caruso et al. [24] also detected a normalization of vitamin D and calcium levels after one or two years with GFD.

Contrary to the results reported in the current evidence, the recommendations for vitamin B12, folates and vitamin C were covered by most of our patients. However, almost none of them met their vitamin D requirements despite having been on GFD for some time. While most of the children covered the recommendations for group B vitamins (B1, B2, B3, B6) and vitamin A, in the case of vitamin E, these recommendations were reached by only a few.

The achievement of this change in dietary habits coincides with the significant increase in complex carbohydrates, fiber, vitamin C and folic acid. The requirements of vitamins D and E continue not to be covered by the majority of the study population. A decrease in vitamin E compliance is observed by avoiding the use of culinary technologies that require a greater amount of oil for their preparation, such as fried and battered ones. The dietary intake of vitamin D increased slightly, but it remained insufficient, so it is necessary to insist on continuing to increase the consumption of oily fish.

With regard to minerals, increasing the consumption of plant-based foods also increased the amounts of magnesium, copper and dietary selenium. However, ingested calcium decreased by 125 mg on a daily basis, increasing by 32.6% the percentage of coeliac patients who did not meet the European recommendations for this mineral. The cause of this decrease in our population lies in the reduction in the consumption of sugary breakfast cereals enriched with micronutrients.

Probably, in order to meet the calcium requirements, it would be necessary to guarantee the intake of 3 servings of dairy products daily, and in turn, have an extra contribution from other sources rich in this mineral, such as fish in which the thorn can be eaten, shellfish, soybeans, chickpeas, almonds, etc.

Finally, it should be noted that a slight decrease of 18.4% was observed in the percentage of patients meeting iron requirements. The possible causes of this non-significant change, together with the fact that half of the patients did not met the requirements of this mineral both at the beginning and at the end of the study, can be several. First, it should be noted that, even tripling the recommended meat consumption, half of the celiac children in the study did not meet iron recommendations. In addition, there are other dietary sources richer in this mineral that are less consumed in the general child population, such as shellfish, especially molluscs and prawns. Lastly, the decrease in the consumption of iron-fortified sugary breakfast cereals is also responsible for this slight decrease.

With these observations, we believe that, instead of recommending increasing the consumption of meat, we should insist on increasing the consumption of fish, shellfish, eggs and nuts to meet the iron recommendations. Taking into account the high sugar content that these products usually have, their consumption should not be recommended to achieve a greater iron and calcium intake.

Studying the relationship of dietary intake by CD patients with their lipid profile, we found some outstanding findings. Significant relationships were found between the consumption of biscuits, the intake of saturated fats and sugars, by children with CD, with the increase in their total blood cholesterol. With processed meats, rich in saturated fat, a relationship was found between their intake with the elevation of LDL cholesterol. Elevated triglycerides were associated with higher consumption of sugary breakfast cereals. However, though the correlations were significant, the correlation coefficients were weak.

It should also be noted that, coinciding with the evidence [62], we found no relationship between dietary cholesterol or “exogenous” cholesterol with the proportion of total cholesterol, LDL cholesterol and HDL cholesterol in the blood.

The present study, together with the review of the literature, highlights the importance of referring all diagnosed CD patients to an RD [63], who focuses on the total intake of their diet, such that, in addition to being gluten-free, it is nutritionally adequate [64] and promotes overall and lifelong health.

Appropriate food and dietetic education for children with CD and their caregivers should focus on a healthy GFD by opting for natural, unprocessed gluten-free cereals [65], which provide adequate nutrients [63], as an alternative to poor-quality packaged foods [66], thereby helping to improve the quality of their diet [48].

However, consumption of processed and ultra-processed foods has increased in the last decades in both developed and developing countries. This disproportionate consumption is adversely impacting general health. A European study found that dietary energy from ultra-processed foods ranked from 10.2% to 50.7%, and its consumption positively correlated with the prevalence of obesity [41]. Moreover, in a Spanish population, a positive association between ultra-processed food consumption and hypertension risk was observed [67], which can be associated to salt content and other lifestyle behaviors.

The negative effects of ultra-processed foodstuffs on children can be even more dangerous, influenced by aggressive marketing. These products, in addition to the high total, saturated and/or trans-fat, free sugars and sodium [68], have additives such as emulsifiers, preservatives, colorants and flavorings that also represent health risks. Because of high glycemic load, the frequent intake of these products can induce obesity and insulin resistance in genetically predisposed children [69]. The eating habits of the pediatric population are increasingly deteriorating. Specifically, Spanish children consume more than 50% of their daily calories in the form of ultra-processed products [70], so it is advisable to undertake strong measures to promote the consumption of healthy foods [71].

Effective nutritional strategies to improve eating habits in children should be a priority to reduce the high prevalence of chronic NCDs in adulthood. Numerous scientific research studies in nutrition end with these recommendations, concluding that prevention education is necessary to improve the current situation of obesity and erratic eating habits, and specifically with CD [72].

However, no studies have been found that evaluate this part fundamental of nutritional treatment in the disease. This is the first RD-initiated study evaluating the efficacy of dietetic education among celiac Spanish children. There are no other comparable studies by which its efficacy can be assessed. All subjects were seen by the same RD, with uniform dietetic advice in all cases. We emphasize the importance of continuous nutritional education as a fundamental part of dietary treatment in these patients.

A major limitation of the present study is that there was no control group. For this reason, it is not possible to know if values at the beginning of the study represent a characteristic of celiac patients or reflect the habits of the general population. Also, participation bias is a possibility because parenteral motivation, socio-economic status, accessibility to fresh produce/gluten-free products, cooking ability, literacy level and number of family members with CD in the household may have impacted on the willingness to take part.

## 5. Conclusions

Our study demonstrated that RD-led nutrition education is able to improve eating patterns in CD pediatric patients. Food and dietetic education aiming to promote a healthy lifestyle should be part of the dietary treatment in order to meet macro- and micro-nutrients requirements. It is also important for improving the quality of diet during childhood, and therefore their future health, in addition to preventing other pathologies associated with a poor diet.

## Figures and Tables

**Figure 1 nutrients-13-01108-f001:**
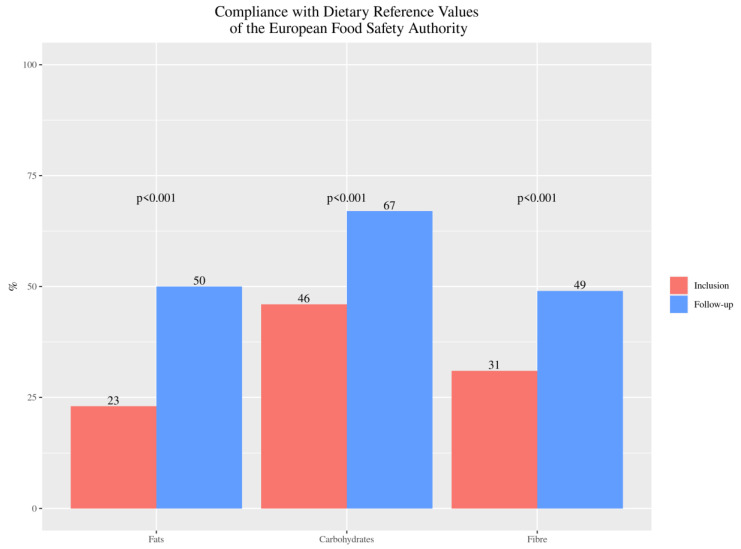
Percentage of celiac disease (CD) pediatric patients’ compliance with the recommendations for fats, carbohydrates (expressed in percentage) and fiber (expressed in g/day) intake, from baseline (inclusion) to twelve months after nutrition education (follow-up).

**Figure 2 nutrients-13-01108-f002:**
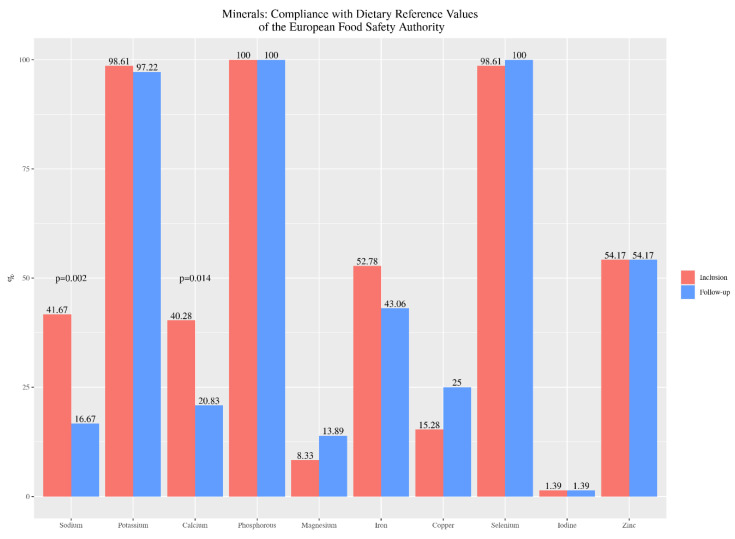
Percentage of celiac children’s compliance with the recommendations for minerals intake, from baseline (inclusion) to twelve months after nutrition education (follow-up).

**Figure 3 nutrients-13-01108-f003:**
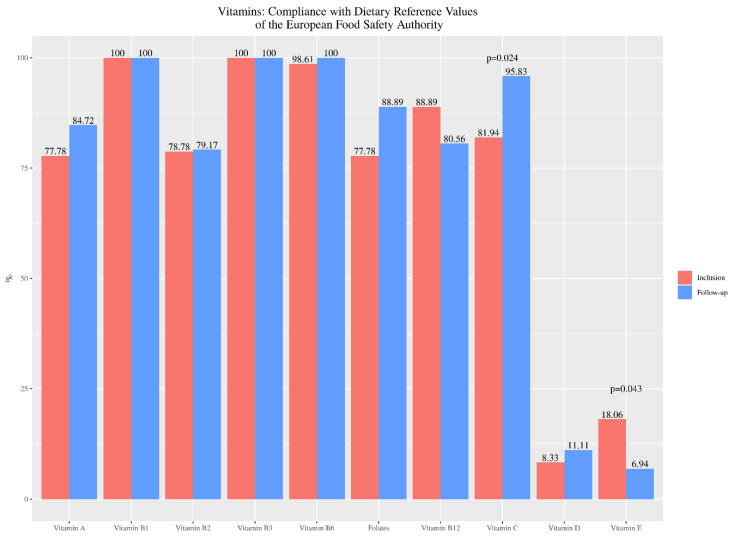
Percentage of celiac children’s compliance with the recommendations for vitamins intake, from baseline (inclusion) to twelve months after nutrition education (follow-up).

**Figure 4 nutrients-13-01108-f004:**
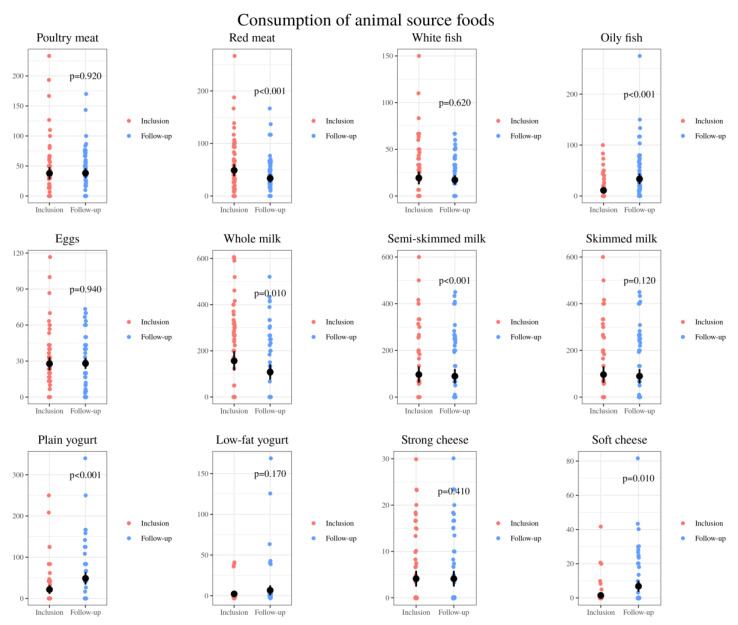
Comparison of the relative intake of animal-based whole foods: grams of poultry meat, red meat, white fish, oily fish, eggs, whole milk, semi-skimmed milk, skimmed milk, plain yogurt, low-fat yogurt, strong cheese, soft cheese per day consumed by 72 CD pediatric individuals at baseline (inclusion) and after nutrition intervention (follow-up).

**Figure 5 nutrients-13-01108-f005:**
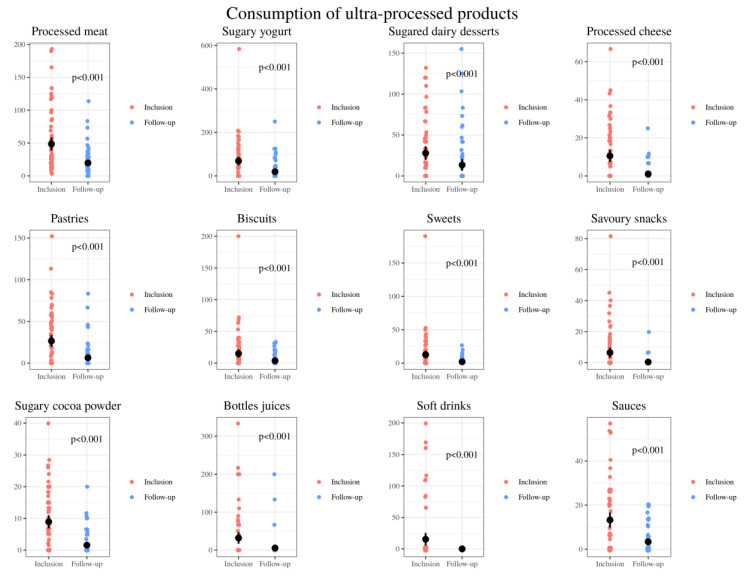
Comparison of the relative consumption of ultra-processed products: grams of processed meat, sugary yogurt, sugared dairy desserts, processed cheese, pastries, biscuits, sweets, savory snacks, sugary cocoa powder, bottles juices, soft drinks and sauces per day consumed by 72 CD pediatric individuals at baseline (inclusion) and after nutrition intervention (follow-up).

**Figure 6 nutrients-13-01108-f006:**
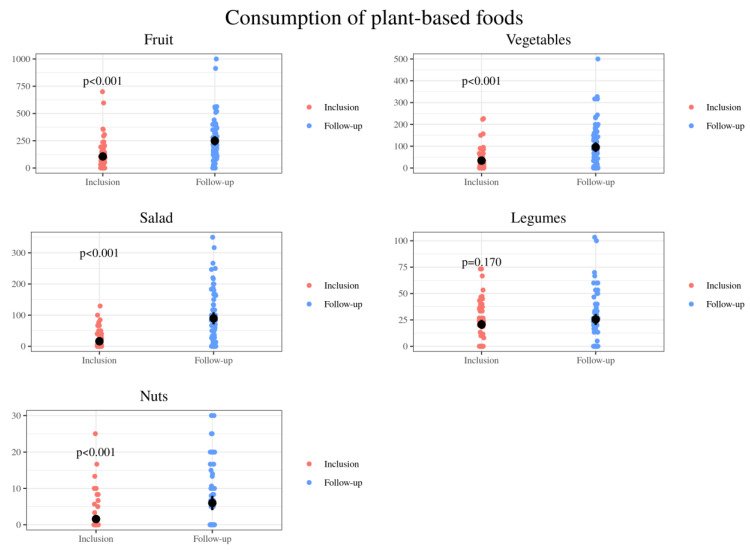
Comparison of the relative consumption of plant-based foods: grams of fruit, vegetables, salad, legumes and nuts per day consumed by 72 CD pediatric individuals at baseline (inclusion) and after nutrition education (follow-up).

**Figure 7 nutrients-13-01108-f007:**
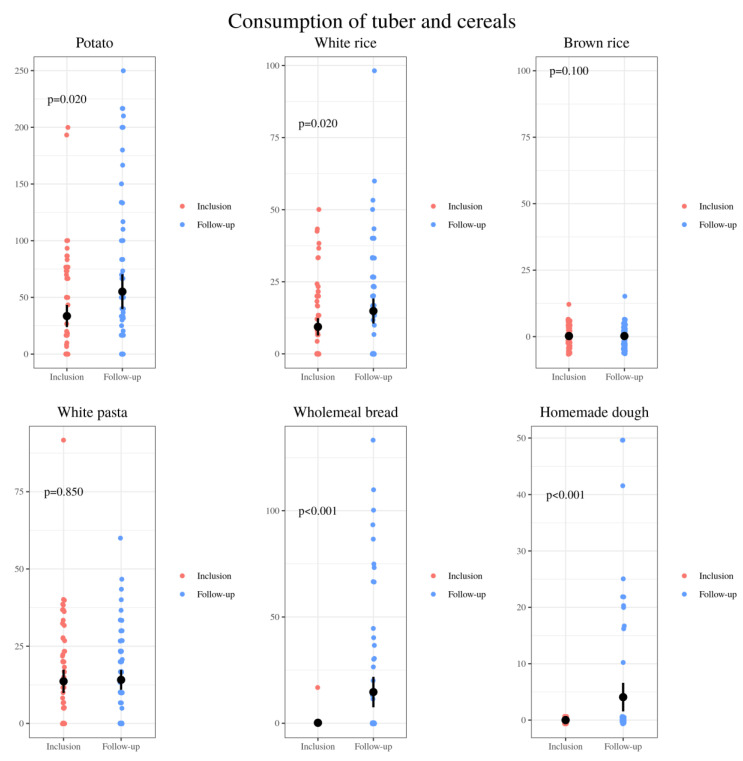
Comparison of the relative consumption of tubers and cereals: grams of potato, white rice, brown rice, white pasta, wheat bread and homemade dough per day consumed by 72 CD pediatric individuals at baseline (inclusion) and after nutrition intervention (follow-up).

**Figure 8 nutrients-13-01108-f008:**
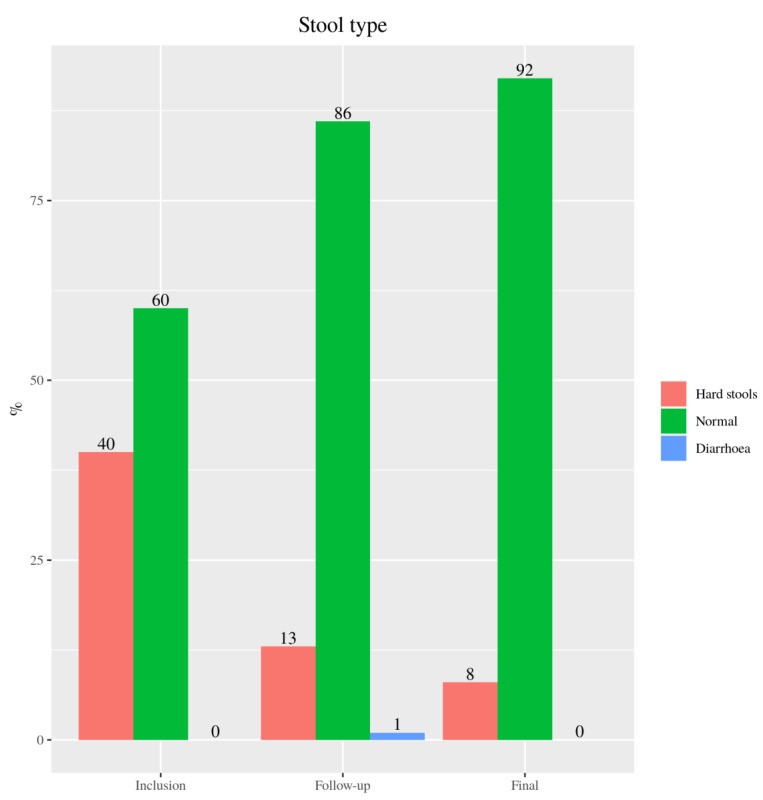
Type of stool data of 72 CD pediatric individuals referred from the baseline (inclusion) and after six (follow-up) and twelve months after the intervention (final).

**Table 1 nutrients-13-01108-t001:** Correlations between lipid profile and dietary intake of 72 celiac children.

Lipid Profile	Dietary Variables	Correlation Coefficient R	*p*-Value	Test
**Total cholesterol**	Biscuits	0.318	0.019	Spearman´s
	Low-fat yogurt	–0.274	0.047	Spearman´s
	Total sugars	0.289	0.034	Spearman´s
	Saturated fat	0.271	0.047	Pearson´s
**HDL cholesterol**	Sugary breakfast cereals	–0.364	0.010	Spearman´s
	Low-fat yogurt	–0.338	0.019	Spearman´s
**LDL cholesterol**	Processed meats	0.365	0.011	Spearman´s
	Saturated fat	0.285	0.050	Pearson´s
	Sugary cocoa powder	0.276	0.058	Spearman´s
**Triglyceride**	Sugary breakfast cereals	0.288	0.039	Spearman´s

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
