# Peer review of "Gluten-Free Diet: Nutritional Strategies to Improve Eating Habits in Children with Celiac Disease: A Prospective, Single-arm Intervention Study"

_nutrients, 2021, doi:10.3390/nu13041108_

Round 1

Reviewer 1 Report

Well written account detailing the changes in gluten free diet for paediatric patients who receive registered dietitian education to improve its general nutritional characteristics . This confirms what has been found previously, that educational intervention by a trained dietitian is extremely important. This study illustrates just how hard it is to do high quality research in this area.

There are some features of the study that I would like to clarify:

Why was the study designed with no control, group? It is possible that families improved over time as they became more familiar with the pitfalls of following a GFD following the diagnosis of coeliac disease. ie was time the main factor or the intervention?

Why had the patients and their families waited so long after a diagnosis without receiving dietetic education about the treatment for their condition?

Also why don't we learn anything about the societal norm for diets for 10 year olds to see how it compared ? matched control data would be ideal, if not consider referencing known data for this age group from the country in which this was conducted.

Deprivation and parental/maternal education can have a major impact on children's diets and GFDs in particular, how did this cohort match up against the general population? Is anything known about these?

Could the authors emphasis the novel features of their study. The discussion needs better referencing from the relevant literature, this aspect need to be carefully considered to how highlight what is novel about this study and their findings.

Reviewer 2 Report

Comments to the authors of the manuscript entitled ”Gluten-Free Diet: Nutritional Strategies to Improve Eating Habits in Children with Coeliac Disease A Prospective, Single-arm Intervention Study” by Gonzalez et al.

The article is well- written and focuses on an important topic. The study lacks a control groups which weakens the results but nicely the authors acknowledge this deficiency in the discussion.

Comments:

  1. Do you have the exact laboratory (vitamins and minerals) values of these patients??
  2. What biochemical values were obtained? Please specify in methods. The exact micronutrient values and the changes of these during the intervention would strengthen the study considerably. Please provide these.
  3. I did not now understand if the micronutrient values were now calculated from the diets or obtained from lab?
  4. Introduction second sentence: should be “Treatment consists OF a lifelong…”?
  5. In the introduction the authors could add that low fibre is related to lower quality of health in CD patients (Laurikka et al. J Clin Gastroenterol 2019 Dietary Factors and Mucosal Immune Response in Celiac Disease Patients Having Persistent Symptoms Despite a Gluten-free Diet)
  6. Also, I would bring up possible consequences of poor diet, especially speculate the risk for atherosclerosis further in life. Now the authors only bring up the deficiencies but do not link them to any clinical risks such as coronary artery disease etc. We do not aim to improve diet per se, but the aim is to prevent diseases further in life through diet.
  7. In results, subsection “Macronutrients” the sentence “After the nutrition intervention the compliance with the recommendations improve in 117.4% for fat intake (p < 0.001), 45.7% for carbohydrates (p < 0.001) and in 58.1% (p < 0.001) for fibre (Fig.1).” The sentence is unclear. Please just state the exact number of patients and percentages of patients who were compliant with the diet before and after. I would remove the percentages as these are misleading in this context as you speak about compliance and not exact laboratory values.
  8. Figure 1. The scale in left should continue until 100% as in all other Figures.
  9. Subsection micronutrients, rows 127-129, you state that “…compliance with magnesium and copper recommendations increased by 66.7% and 63.6%…” I would again ask you to present the exact number of patients and percentage of all patients that were more compliant and the change in a similar fashion. See comment 4.
  10. Subsection 3.3. Same as in comment 7.
  11. Table 1. The correlation coefficients are very low indicating a low correlation. P-values are mostly useless in correlation statistics. The authors should clearly state that found only WEAK correlations between lipid profiles and dietary intakes. In the methods should be stated what are considered strong (usually above 0.7) moderate (usually 0.4-0.6) and weak (usually 0.2-0.4) correlation co-efficients. (ref. Evans, J. D. (1996). Straightforward statistics for the behavioral sciences. Pacific Grove, CA: Brooks/Cole)
  12. In discussion, rows 281-289: should be changed according to comment 8. Though the correlations were significant, the correlation coeffiecients were weak.
  13. In the new paper by Dotsenko et al. it was stated that even in CD patients with normal mucosa there are changes in the RNA of certain transporters handling micronutrient transportation so perhaps CD patients need different recommendations than other patients? Maybe it is not a problem of the GFD but the disease? Please speculate. (Dotsenko et al. Genome-Wide Transcriptomic Analysis of Intestinal Mucosa in Celiac Disease Patients on a Gluten-Free Diet and Postgluten Challenge. Cell Mol Gastroenterol Hepatol. 2021; 11(1): 13–32.)
  14. In the discussion authors state the nutrient requirements but do not state what happens if these are low. The authors should state WHY it is important to follow these recommendations. What happens if celiac patient does not follow these RD recommendations?? In conclusions it is stated that meeting these nutritional requirements prevents pathologies, but none such statements are made in introduction and discussion? Please provide this info.
  15. In the abstract you provide only p-values, please add the number of patients and percentage of patients before and after the diet who were compliant with the diet (same as comment 7).
  16. The figures 4-7 are very nice.
  17. The discussion has too many chapters, it is difficult to follow as the story jumps from one to another so frequently. Please combine some of the chapters.

Round 2

Reviewer 2 Report

Dear González et al,

I Comment your second revision of the manuscript entitled “Gluten-Free Diet: Nutritional Strategies to Improve Eating Habits in Children with Coeliac Disease A Prospective, Single-arm Intervention Study”.

I'm very satisfied with your changes and answers. The manuscript is overall more interesting and the science is coherent. Therewith, I gladly suggest to accept your manuscript.

Author Response

Thank you very much indeed